# Morphological Study of the Mandibular Lingula and Antilingula by Cone-Beam Computed Tomography

**DOI:** 10.3390/bioengineering10020170

**Published:** 2023-01-28

**Authors:** Chun-Ming Chen, Hui-Na Lee, Shih-Wei Liang, Kun-Jung Hsu

**Affiliations:** 1School of Dentistry, College of Dental Medicine, Kaohsiung Medical University, Kaohsiung 80708, Taiwan; 2Department of Oral and Maxillofacial Surgery, Kaohsiung Medical University Hospital, Kaohsiung 80756, Taiwan; 3Division of Conservative Dentistry, Kaohsiung Medical University Hospital, Kaohsiung 80756, Taiwan; 4Graduate Institute of Dental Sciences, College of Dental Medicine, Kaohsiung Medical University, Kaohsiung 80708, Taiwan; 5Department of Dentistry, Kaohsiung Medical University Hospital, Kaohsiung 80756, Taiwan

**Keywords:** lingula, antilingula, mandibular foramen, presence probability, sagittal split ramus osteotomy, intraoral vertical ramus osteotomy

## Abstract

The patterns of the lingula and antilingula are crucial surgical reference points for ramus osteotomy. Cone-beam computed tomography (CBCT) provides three-dimensional images, and patient radiation dose is significantly lower for CBCT than for medical CT. The morphology of the mandibular lingula and antilingula of ninety patients (180 sides) were investigated using CBCT. The lingula were classified as having triangular, truncated, nodular, and assimilated shapes. The antilingula were classified as having hill, ridge, plateau, and plain shapes. The patients’ sex, skeletal patterns (Classes I, II, and III), and right and left sides were recorded. The most to least common lingula shapes were nodular (37.8%), followed by truncated (32.8%), triangular (24.4%), and assimilated (5%). The most to least common antilingulae were hill (62.8%), plain (18.9%), plateau (13.9%), and ridge (4.4%) patterns, respectively. The lingula and antilingula had identical patterns on both sides in 47 (52.2%) and 46 patients (51.1%), respectively. Sex and skeletal pattern were not significantly correlated to lingula and antilingula shapes. No significant correlation was observed between lingula and antilingula shapes.

## 1. Introduction

The mandibular foramen is an opening on the inner surface of the mandibular ramus. It provides a passage for the inferior alveolar nerve and blood vessels to enter the mandibular canal [1]. The lingula is a tongue-shaped bony projection superior to the mandibular foramen, and the antilingula is at the outer surface of the mandibular ramus, opposite to the lingula, and typically refers to a bony prominence [2]. Passing through the orifice of the mandibular foramen, ramus, and mandibular body and entering the lower lip area through the mental foramen, the inferior alveolar nerve functions as the sensory nerve of the mandibular teeth and the skin of the lower lip and chin. Therefore, the mandibular foramen, lingula, and antilingula are closely associated with the position of the inferior alveolar nerve and blood vessels [3].

The relative positions of the mandibular foramen, lingula, and antilingula are crucial, particularly during mandibular setback surgery for patients with mandibular prognathism [4,5,6,7]. During ramus surgery, surgeons must identify the positions of the landmarks (lingula and antilingula) and avoid operating at the mandibular foramen lest they damage the inferior alveolar nerve or blood vessels, which can lead to intraoperative hemorrhage or postoperative lower lip paresthesia or numbness. Surgical landmarks (lingula and antilingula) may present with different morphologies and may sometimes be absent [8,9,10,11,12,13,14].

In India, Tuli et al. [9] first classified the lingula in the adult dry mandible into four shapes: triangular, truncated, nodular, and assimilated shapes, which have prevalence rates of 68.5%, 15.8%, 10.9%, and 4.8%, respectively. CBCT has a low-dose radiation and provides accurate three-dimensional (3D) images of the craniofacial bones. It has been commonly used as 3D cephalometric analysis for orthodontic diagnosis and treatment [15,16]. In Turkey, Sekerci and Sisman [17] identified morphological shapes of the mandibular lingula using CBCT. They reported that nodular, truncated, triangular, and assimilated shapes, which accounted for 51.2%, 32.0%, 14.1%, and 2.7%, respectively. However, no report has discussed the shapes of the antilingula. Therefore, in this study, we explored the shapes and occurrence of the lingula and antilingula in the mandible. Analyses were conducted with respect to sex, three skeletal classes, and the left and right sides of the ramus to explore correlations between the lingula and antilingula. Our epidemiological findings can guide treatment strategies and surgical planning to prevent neurovascular bundle injury during mandibular ramus surgery.

## 2. Materials and Methods

### 2.1. Data Conditions

Cone-beam computed tomography (CBCT) images were collected from the Department of Dentistry at Kaohsiung Medical University Hospital. During CBCT (New Tom VGi Evo, Imola, Italy) scan, patients were under natural head position with centric occlusion. The sizes of the field of view (FOV) were 24 × 19 cm. The CBCT was operated at 110 kV, 4.59 mA, and X-ray emission time was 3.5 s with a voxel of 0.03 mm. The inclusion criteria were as follows: (1) all patients were Taiwanese; (2) the age of each patient was 16 years old or older; (3) mandibles were dentate and included the second molar; and (4) CBCT image showed the good contrast resolution quality for the classifications of the lingula and antilingula. The exclusion criteria were as follows: (1) patient age under 16 years, (2) mandibular pathology, (3) a history of orthognathic surgery, and (4) mandibular trauma. A total of 90 patients (60 women and 30 men) were categorized into three classes according to skeletal patterns (30 patients per class). The mean age was 25.3 years (range: 16–46 years). Specifically, the Point A–Nasion–Point B (ANB) angle of 0°–4°, >4°, and <0° corresponded to Class I (mean age, 25.3 years; 21 women and 9 men), Class II (mean age, 27.8 years; 24 women and 6 men), and Class III (mean age, 22.9 years; 15 women and 15 men), respectively. We applied G∗Power version 3.1.9.2 (Franz, Universitat Kiel, Kiel, Germany) to calculate sample size. After specifying α = 0.05, we recruited 90 patients to achieve sufficient power of 83%. During 3D cephalometry on CBCT data, inter-operator variability was examined. The percent agreement (81/90) was 90 with a Kappa value of 0.85.

### 2.2. Image Investigation and Analysis

All CBCT images (DICOM files) were imported into RadiAnt DICOM Viewer (version 4.6.9, Medixant, Poznan, Poland) software to reconstruct and segment the 3D image. After extraction of the mandibular ramus (separated from the maxilla), four shapes of the lingula (Figure 1: triangular, truncated, nodular, and assimilated) [9] were observed. The antilingula patterns (Figure 2) were divided into four shapes of landforms (hill, ridge, plateau, and plain). The absence of lingula and absence of the antilingula were defined as assimilated and plain shapes, respectively. The three skeletal patterns were analyzed by sex and by side (left and right).

### 2.3. Statistical Analysis

The chi-square test and McNemar–Bowker test were used to conduct statistical analysis. Data were analyzed using IBM SPSS 20 (SPSS, Chicago, IL, USA), and *p* < 0.05 indicated statistical significance. The null hypothesis stipulated a nonsignificant correlation in between any lingula and antilingula shape and sex or skeletal patterns.

## 3. Results

### 3.1. Morphological Parameters of Sex and Skeletal Patterns

Among the 90 patients (180 sides), the shapes of the lingula (Table 1) were triangular in 24.4% (44 sides), truncated in 32.8% (59 sides), nodular in 37.8% (68 sides), and assimilated in 5% (9 sides), respectively (Table 1). The nodular shape was most common among women, and the truncated shape was most common among men. Moreover, the truncated shape was most common in Class I, and the nodular shape was most common in Class II and III. The shapes of the antilingula were hill shape in 62.8% (113 sides), ridge shape in 4.4% (8 sides), plateau shape in 13.9% (25 sides), and plain shape in 18.9% (34 sides). The hill shape was most common among both women and men as well as among all skeletal patterns. Thus, sex or skeletal pattern was not significantly correlated with lingula or antilingula shape. The null hypothesis was accepted.

### 3.2. Morphological Parameters on the Right and Left Sides for Each Sex

Table 2 indicates whether the lingula and antilingula were present on the left and right sides of the ramus. Nodular and truncated shapes of the lingula were the most common on the right (37 sides) and left (40 sides) sides, respectively, and the assimilated shape was the least common on the right (7 sides) and left (2 sides) sides. In women, nodular and truncated shapes of the lingula were the most common on the right (28 sides) and left (27 sides) sides. The hill shape of the antilingula was the most common on the right (65 sides) and left (48 sides) sides. Ridge and plateau shapes of the antilingula were the least common on the right (5 sides) and left (2 sides) sides. The McNemar–Bowker test revealed a significant difference in lingula and antilingula shapes between men and women.

### 3.3. Morphological Parameters in the Bilateral and Unilateral Sides

Table 3 indicates whether the lingula and antilingula exhibited identical shapes on the left and right sides of the ramus. Identical lingula shape on both sides was present in 94 sides, and the nodular shape (40 sides) was the most common. Identical antilingula shape on both sides was present 92 sides, and the hill shape (76 sides) was the most common.

### 3.4. Morphological Parameters on the Right and Left Sides for Skeletal Patterns

As indicated in Table 4, for the three skeletal patterns, the truncated shape of the lingula was the most common on the left side for Class I. The nodular shape of the lingula was the most common on both sides for Class II and III, and the hill shape of the antilingula was the most common in the right and left sides for all skeletal patterns.

### 3.5. Morphological Parameters in the Bilateral and Unilateral Sides of the Skeletal Patterns

As indicated in Table 5, for the three skeletal patterns, an identical lingula shape on both sides was noted in 28 sides for Class I (most common: truncated), 38 sides for Class II (most common: nodular), and 28 sides for Class III (most common: triangular). With respect to antilingula shapes, the same shape was observed on both sides for Class I (36 sides), Class II (34 sides), and Class III (22 sides).

### 3.6. Distribution Correlation between the Lingula and Antilingula Shapes

As noted in Table 6, no significant correlation was noted between lingula and antilingula shapes. The hill and ridge shapes were the most and least common shapes of the lingula, respectively. Reviewing the research literature, demographic characteristics and lingual shapes are shown in Table 7. In the literature review, there is no repost about antilingula shape.

## 4. Discussion

The edge of most mandibular foramens has an irregular surface, and there is a protruding ridge anterosuperior to the edge which forms a bone spine to cover the orifice of the mandibular foramen. This bone spine is called a lingula. The sphenomandibular ligament, which is attached to the mandibular foramen orifice and the lingula, restricts the anteroinferior movement of the mandible and protects the inferior alveolar neurovascular bundle from being directly exposed before entering the mandibular foramen. Therefore, lingula shape is also affected by the attachment and strength of the sphenomandibular ligament. However, considerable variations have been reported in the occurrence and distribution of lingula shapes. Murlimanju et al. [19] reported that the triangular, truncated, nodular, and assimilated shapes were present in 29.9%, 27.6%, 29.9%, and 12.6% of cases, respectively. Jansisyanont et al. [6] reported them to be present in 29.9%, 46.2%, 19.6%, and 4.3% of cases, respectively; and Jung et al. [13] reported them to be present in 14.3%, 29.3%, 54.0%, and 2.4% of cases, respectively. These discrepancies may be attributed to differences in the age of the sample population, ethnicity, dentition, and skeletal patterns, and the process of making and preserving the human mandibular specimen. Our results are similar to those of Jung et al. [13]. The truncated and nodular lingula shapes appeared to be the most common, and the assimilated shape was the least common.

In our study, the same lingula shape on both left and right sides was observed in 52.2% of sides. In 51.7% of women and 46.7% of men, the same lingula shape was present on both sides. Jung et al. [13] reported that 74.7% and 60.5% of individuals with Class I and Class III shapes, respectively, had the same lingula shape on both sides, and the difference was not statistically significant. In our study, 63.3% of individuals with the Class II shape had the same lingula on the left and right sides, which was higher than the value for Class I (46.7%) and Class III (46.7%). Our results revealed no significant difference in the symmetry of the bone shapes compared to the values reported by Jung et al. [13]. Furthermore, our data indicated that most men and women had truncated and nodular shape lingula, and the difference was not significant. In our study, the truncated shape was the most common in Class I lingula, whereas the nodular shape was the most common in Class II and III lingula. Jung et al. [13] reported that the lingula does not significantly differ between sexes and skeletal patterns (Class I and III). Our results are consistent with those of Jung et al. [13].

The mandibular lingula is an important anatomical landmark that is often used clinically to approximate the location of the mandibular foramen. Choi and Hur [24] reviewed the mandibular-lingula-related literature and evaluated the efficiency in the inferior alveolar nerve block. They reported that most lingula are located superior to the occlusal plane and that the triangular shape was generally located slightly more superiorly and posteriorly than other shapes. Furthermore, the lingula is located more posteriorly and superiorly in the prognathic mandible than in the nonprognathic mandible. Zhou et al. [25] evaluated the mandibular lingula and foramen location using 3-dimensional mandibular models reconstructed by CBCT. They reported that the mandibular lingula was rarely located inferior to the occlusal plane; however, the position of the mandibular foramen was more variable (84.3% inferior to, 12.4% superior to, and 3.3% at the level of the occlusal plane). Lupi et al. [26] investigated the anatomical relationship between the lingula and anatomical measurements using CBCT. They recommended that dentists or surgeons insert a needle approximately 16.96 mm from the anterior border of the ramus and approximately 11.22 mm superior to the occlusal plane.

Lima et al. [27] investigated the locational, shape-based, and anatomical relations of the mandibular foramen and the mandibular lingula for surgical procedures in the mandibular ramus. They reported that the distance from the mandibular foramen to the mandibular lingula was 7.88 ± 2.15 mm on the right side and 7.77 ± 2.01 mm on the left side of the mandible. The distance from the lingula to the mandibular notch was 21.35 ± 3.59 mm on the right side and 21.05 ± 3.09 mm on the left side. Samanta and Kharb [28] performed a morphological analysis of the lingula in dry adult human mandibles of a north Indian population. The most common shape was the triangular shape, and the height of the lingula was 5.5 ± 2.0 mm. They reported that the lingula from the anterior borders of the ramus and mandibular notch were 20.0 ± 2.4 mm and 15.4 ± 2.7 mm, respectively. Woo et al. [29] investigated 65 Korean human dry mandibles to measure the distances between the lingula and mandibular foramen. They reported that the anterior ramal horizontal distance to the lingula and to the mandibular foramen was 16.13 ± 3.53 mm and 23.91 ± 4.79 mm, respectively. The vertical distance from the sigmoid notch to the lingula was 19.82 ± 5.11 mm, and the height of the lingula was 10.51 ± 3.84 mm. Consequently, the locations of osteotomy lines and amount of periosteal elevation make injury of the neurovascular bundle unlikely during mandibular split operation.

Currently, the two most common surgical methods for mandibular setback are sagittal split ramus osteotomy (SSRO) and intraoral vertical ramus osteotomy (IVRO). SSRO is performed on the inner side of the mandibular ramus. The osteotomy must be superior to the mandibular foramen to avoid damage to the inferior alveolar neurovascular bundle, which can cause massive bleeding during an operation and neurosensory disturbance in the lower lip after an operation. Thus, surgeons must fully understand the anatomy surrounding the mandibular foramen. The IVRO operation is performed from the outer side of the mandibular ramus, and the osteotomy must be behind the mandibular foramen to avoid injury to the inferior alveolar neurovascular bundle. The antilingula is usually used as a reference for the operation. The term “antilingula” was first defined by Levine and Topazian [30] in 1976. It was used as a reference for inverted-L osteotomy. At that time, it was believed that the antilingula was formed by the inferior alveolar nerve entering the mandibular ramus, which caused the formation of a prominent bulge in the outer surface. Therefore, Levine and Topazian [30] suggested that the position of the antilingula could be used as a landmark reference for IVRO surgery. However, the relationship between the antilingula, the lingula, and the mandibular foramen is not fixed. Monnazzi et al. [1] assessed their anatomical locations in dry mandibles and reported that the mandibular foramen was on average 5.82 mm inferior to the lingula. Park et al. [31] indicated that the antilingula is located 4.98 mm anterior to and 6.95 mm superior to the mandibular foramen. Aziz et al. [5] reported that the lingula was found anterior to the antilingula in 33% of specimens and posterior to the antilingula in 45.6%. The lingula was superior to the antilingula in 47.2% of the specimens and inferior in 50%. Given the no fixed relationship between the lingula and the antilingula, Monnazzi et al. [1] did not recommend the use of the antilingula as a landmark for vertical ramus osteotomy.

Reitzik et al. [32] later described the antilingula as a masseteric apical bump. Further studies on humans and other mammals revealed that the bony protuberance on the outer side of the ramus is the attachment area of the deep head of the masseter muscle [33]. Accordingly, not every mandible has a distinct antilingula. Therefore, the shape of the antilingula may be affected by attachment force and area of the masseter muscle. We believe the opinion of Reitzik et al. [32] to be more plausible. Because the deep head of the masseter muscle has a wider attachment area and greater strength, similar to the uplift of the terrain, we classified the antilingula into four shapes of landforms based on the observations of clinical patients and CBCTs into hill, ridge, plateau, and plain shapes. In our study, only 62.8% of the antilingula (hill shape) could be clearly used as a reference point for IVRO surgery, and in 18.9% of cases, the plain shape (without antilingula) was observed.

The antilingula had the same shape on both sides in 51.1% sides. Moreover, 50% of the women had the same antilingula on both sides, whereas 53.3% of men exhibited this feature. The antilingula was the same on both sides in Class I (60%) and Class II (56.7%), the proportions of which were much higher than those of Class III (36.7%). This indicates that when IVRO was used on 63.3% of sides, the antilingula shape on both sides differed; therefore, we should be careful when judging the position of the antilingula to avoid the situation in which osteotomy does not occur on the mandibular foramen, causing injury to the inferior alveolar neurovascular bundle. Most men and women had truncated and nodular shape lingulae and only hill-shape antilingulae, with no significant between-sex differences. The antilingulae of Classes I, II, and III were mostly of the hill shape, with no significant between-group differences. We observed that sex and skeletal patterns were not significantly correlated with the shapes of the lingula and antilingula.

## 5. Conclusions

The clinical significance of the lingula and antilingula shapes is that they act as important surgical landmarks for locating the mandibular foramen and, thus, avoiding damage to the neurovascular bundle during ramus osteotomy. With the recent advances in biomaterial and 3D image simulation, 3D planning software and printing technology (rapid prototyping, personalized orthognathic surgical guide, miniplate fixation devise, etc.) provide the virtual and precise osteotomy at the lingula and antilingula locations of ramus surgery. Nodular and hill shapes are the most common for the lingula and antilingula, respectively. Sex and skeletal patterns had no significant correlation with the lingula and antilingula.

## Figures and Tables

**Figure 1 bioengineering-10-00170-f001:**
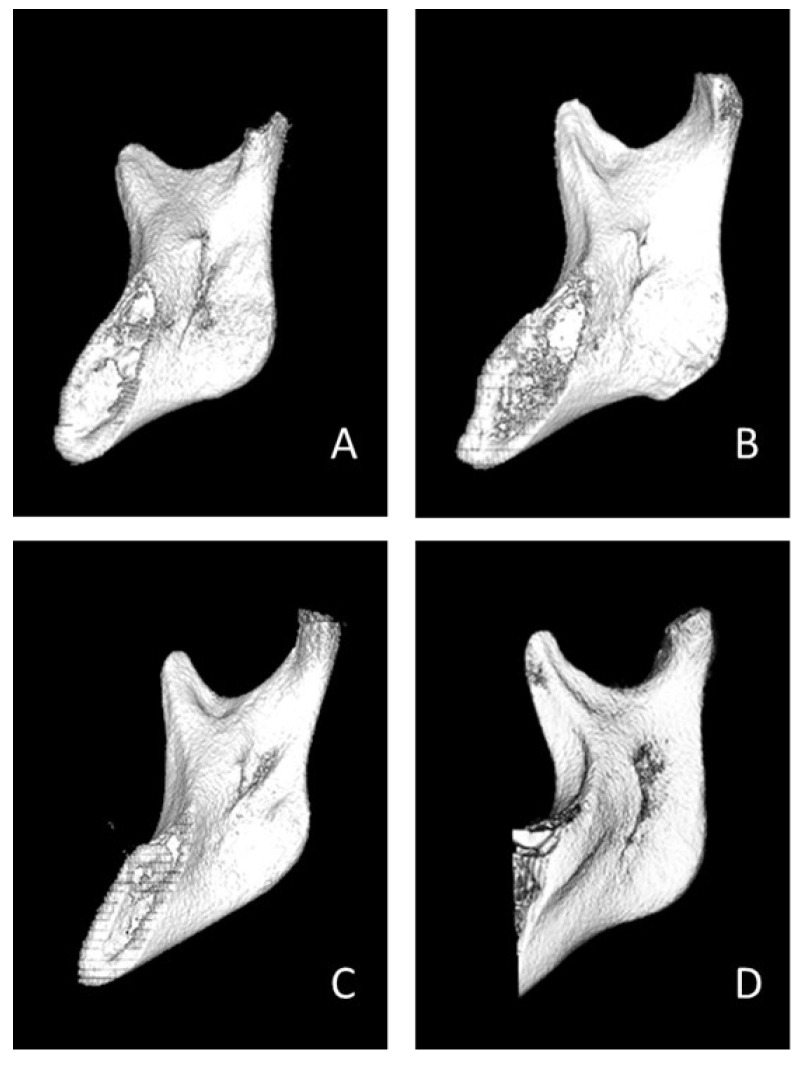
In the medial aspect of right mandibular ramus, four shapes of lingula were classified. (**A**) Triangular, (**B**) Truncated, (**C**) Nodular, and (**D**) Assimilated.

**Figure 2 bioengineering-10-00170-f002:**
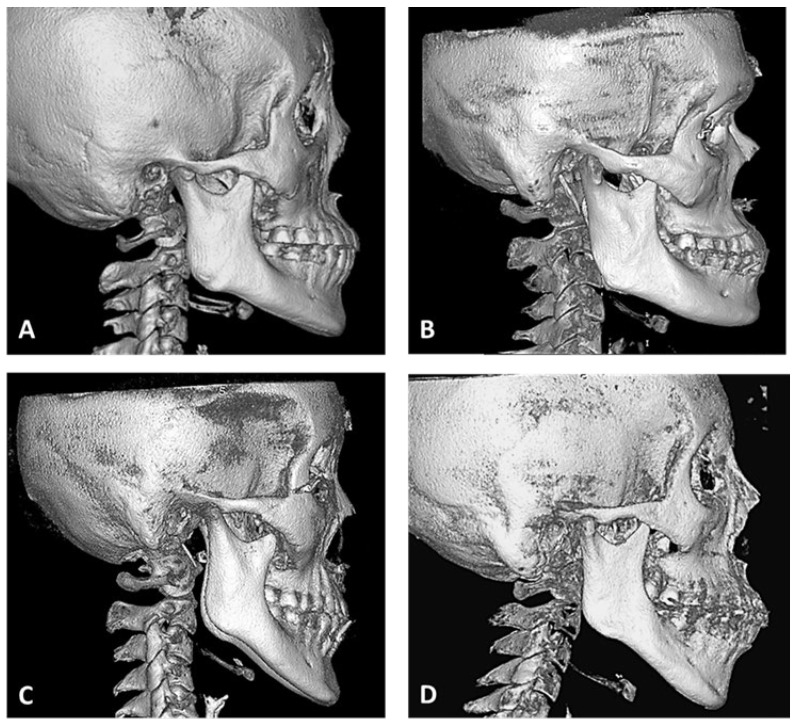
In the lateral aspect of right mandibular ramus, four shapes of antilingula were classified. (**A**) Hill, (**B**) Ridge, (**C**) Plateau, and (**D**) Plain.

**Table 1 bioengineering-10-00170-t001:** Distribution of lingula and antilingula shapes in mandible.

Shape	Gender						Skeletal Pattern							
	Total (n = 180)		Male (n = 60)		Female (n = 120)		Chi-Square *p* Value	Class I (n = 60)		Class II (n = 60)		Class III (n= 60)		Chi-Square *p* Value
Lingula														
Triangular	44	24.4%	13	7.2%	31	17.2%	0.366	17	9.4%	10	5.6%	17	9.4%	0.232
Truncated	59	32.8%	22	12.2%	37	20.6%		24	13.3%	17	9.4%	18	10.0%	
Nodular	68	37.8%	20	11.1%	48	26.7%		16	8.9%	29	16.1%	23	12.8%	
Assimilated	9	5.0%	5	2.8%	4	2.2%		3	1.7%	4	2.2%	2	1.1%	
Total	180	100.0%	60	33.3%	120	66.7%		60	33.3%	60	33.3%	60	33.3%	
Antilingula														
Hill	113	62.8%	43	23.9%	70	38.9%	0.101	39	21.7%	35	19.4%	39	21.7%	0.200
Ridge	8	4.4%	4	2.2%	4	2.2%		5	2.8%	1	0.6%	2	1.1%	
Plateau	25	13.9%	4	2.2%	21	11.7%		10	5.6%	8	4.4%	7	3.9%	
Plain	34	18.9%	9	5.0%	25	13.9%		6	3.3%	16	8.9%	12	6.7%	
Total	180	100.0%	60	33.3%	120	66.7%		60	33.3%	60	33.3%	60	33.3%	

n: number of sides.

**Table 2 bioengineering-10-00170-t002:** Distribution of lingula and antilingula shapes in the right and left sides by gender.

Shape	Total (n = 180)		McNemar–	Male (n = 60)		Female (n = 120)	
	Right	Left	Bowker Test	Right	Left	Right	Left
Lingula			*p* < 0.001 *				
Triangular	27	17		8	5	19	12
Truncated	19	40		9	13	10	27
Nodular	37	31		9	11	28	20
Assimilated	7	2		4	1	3	1
Total	90	90		30	30	60	60
Antilingula			*p* = 0.047 *				
Hill	65	48		25	18	40	30
Ridge	5	17		1	3	4	14
Plateau	9	2		2	2	7	0
Plain	11	23		2	7	9	16
Total	90	90		30	30	60	60

n: number of sides; *: Statistically significant, *p* < 0.05.

**Table 3 bioengineering-10-00170-t003:** Distribution of lingula and antilingula shapes in the bilateral and unilateral sides of gender.

Shape	Total (n = 180)		Male (n = 60)		Female (n = 120)	
	Bilateral	Unilateral	Bilateral	Unilateral	Bilateral	Unilateral
Lingula						
Triangular	24	20	6	7	18	13
Truncated	28	31	14	8	14	23
Nodular	40	28	12	8	28	20
Assimilated	2	7	0	5	2	2
Total	94	86	32	28	62	58
Antilingula						
Hill	76	37	30	13	46	24
Ridge	2	6	2	2	0	4
Plateau	4	43	0	4	4	17
Plain	10	45	0	9	10	15
Total	92	88	32	28	60	60

n: number of sides.

**Table 4 bioengineering-10-00170-t004:** Distribution of lingula and antilingula shapes in the right and left sides of skeletal patterns.

Shape	Class I (n = 60)		Class II (n = 60)		Class III (n = 60)	
	Right	Left	Right	Left	Right	Left
Lingula						
Triangular	8	7	8	2	9	8
Truncated	9	17	5	12	7	11
Nodular	9	5	14	15	12	11
Assimilated	4	1	3	1	2	0
Total	30	30	30	30	30	30
Antilingula						
Hill	21	18	21	14	23	16
Ridge	4	1	0	1	1	1
Plateau	4	6	3	5	2	5
Plain	1	5	6	10	4	8
Total	30	30	30	30	30	30

n: number of sides.

**Table 5 bioengineering-10-00170-t005:** Distribution of lingula and antilingula shapes in the bilateral and unilateral sides of skeletal patterns.

Shape	Class I (n = 60)		Class II (n = 60)		Class III (n = 60)	
	Bilateral	Unilateral	Bilateral	Unilateral	Bilateral	Unilateral
Lingula						
Triangular	8	9	4	6	12	5
Truncated	14	10	8	9	6	12
Nodular	6	10	24	5	10	13
Assimilated	0	3	2	2	0	2
Total	28	32	38	22	28	32
Antilingula						
Hill	30	9	24	11	22	17
Ridge	2	3	0	1	0	2
Plateau	2	8	2	6	0	7
Plain	2	4	8	8	0	12
Total	36	24	34	26	22	38

n: number of sides.

**Table 6 bioengineering-10-00170-t006:** Distribution correlation between lingula and antilingula shapes.

	Lingula (n)					Chi-Square
Shape	Triangular	Truncated	Nodular	Assimilated	Total	*p* Value
Antilingula (n)						
Hill	22	39	45	7	113	0.197
Ridge	4	1	3	0	8	
Plateau	10	10	5	0	25	
lain	8	9	15	2	34	
Total	44	59	68	9	180	

n: number of sides.

**Table 7 bioengineering-10-00170-t007:** Demographic characteristics and shapes of lingula (dry mandible and Cone-beam computed tomography: CBCT) in literature.

Author	Material	Patients (Sides)	Age (years)	Shape of Lingula (%)			
Year, Country		Female (F)/Male (M)	Mean, Range	Triangular	Truncated	Nodular	Assimilated
Tuli et al. [9]	Dry	n = 165 (330)	NA	68.5%	15.8%	10.9%	4.8%
2000, India	mandible	34 F/131 M					
Kositbowornchai et al. [18]	Dry	n = 72 (144)	27–87 years	16.66%	47.22%	22.92%	13.19%
2007, Thailand	mandible	20 F/52 M					
Jansisyanont et al. [6]	Dry	n = 92 (184)	42.4	29.9%	46.2%	19.6%	4.3%
2009, Thailand	mandible	34 F/58 M	18–83				
Murlimanju et al. [19]	Dry	n = 67 (134)	Adult	29.9%	27.6%	29.9%	12.6%
2012, India	mandible	30 F/37 M					
Sekerci and Sisman [17]	CBCT	n = 412 (824)	Adult	14.1%	32.0%	51.2%	2.7%
2014, Turkey		199 F/213 M					
Senel et al. [12]	CBCT	n = 63 (126)	46	22.2%	19.0%	32.5%	26.2%
2015, Turkey		28 F/35 M	25–70				
Alves and Deana [20]	Dry	n = 132 (253)	Adult	23.3%	49.0%	26.5%	1.2%
2016, Brazil	mandible						
Asdullah et al. [21]	Dry	n = 50 (100)	Adult	42.0%	32.0%	20.0%	6.0%
2018, India	mandible	25 F/25 M					
Jung et al. [13]	CBCT	n = 347 (694)	27	14.3%	29.3%	54.0%	2.4%
2018, Korea		166 F/181 M	19–50				
Akcay et al. [7]	CBCT	n = 60 (120)	18–37	20.0%	21.7%	45.0%	13.3%
2019, Turkey		30 F/30 M					
Ahn et al. [22]	CBCT	n = 30 (58)	NA	31.0%	25.9%	32.8%	10.3%
2020, Korea							
Stipo et al. [23]	Dry	n = 235 (453)	69.4	10.8%	38.6%	26.3%	4.0% *
2022, Italy	mandible	114 F/121 M	20–101				
Present study	CBCT	n = 90 (180)	25.3	24.4%	32.8%	37.8%	5.0%
2023, Taiwan		60 F/30 M	16–46				

n: number of patients; NA: Not available; *: Assimilated: 4.0%, Mixed: 15.2%, Bridge: 5.1%.

## Data Availability

The data used to support the findings of this study are available from the corresponding author upon request.

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
