# Peer review of "Morphological Study of the Mandibular Lingula and Antilingula by Cone-Beam Computed Tomography"

_bioengineering, 2023, doi:10.3390/bioengineering10020170_

Round 1

Reviewer 1 Report

Congratulations to the authors for compiling a good study. 

Some points to consider before the paper is accepted 

Abstract:

“This study investigated the morphology of mandibular lingula and antilingula. Ninety 19 patients (180 sides) underwent morphological investigations…”

Were the scans taken specially for the current investigation ? If so how do the authors justify the exposure to patients.?

Introduction:

The authors should mention about any previous literature regarding any population wherein the anatomy was reported. Further more they can compare their findings to the current investigation and mention it in the discussion.

Materials and Methods:

The authors fail to mention how the sample size was calculated. Also have a sample size as 90 is very less for a conclusion they draw. Please justify.

The parameters with which the CBCT scanning was done is missing in the methodology. Please add it. The POV, Exposure time, etc to be mentioned is very important here.

The results and discussion is ok.

In Discussion it is suggested to the authors to add 2 review tables of previous studies in different populations and with the current investigation. 

The review table can have the Authors, Year, Population, sample size, method use for evaluation (CBCT, OPG, Lat ceph, any other), dimensions

Thank you 

Author Response

  1. Abstract:

“This study investigated the morphology of mandibular lingula and antilingula. Ninety  patients (180 sides) underwent morphological investigations…”

Were the scans taken specially for the current investigation? If so how do the authors justify the exposure to patients?

Answer: Cone-beam computed tomography (CBCT) provides three-dimensional image and patient radiation dose is significant lower for CBCT than for medical CT. Ninety patients (180 sides) were investigated the morphology of mandibular lingula and antilingula using CBCT. The lingula were classified as triangular, truncated, nodular, and assimilated type. The antilingula were classified as hill, ridge, plateau, and plain type.

  1. Introduction:

The authors should mention about any previous literature regarding any population wherein the anatomy was reported. Further more they can compare their findings to the current investigation and mention it in the discussion.

Answer: We add the following sentences at the third paragraph of Introduction.

  • In India, Tuli et al. [9] firstly classified the lingula in the adult dry mandible into 4 shapes: triangular, truncated, nodular and assimilated shapes, which have prevalence rates of 68.5%, 15.8%, 10.9%, and 4.8%, respectively.
  • In Turkey, Sekerci and Sisman [17] identified morphological shapes of the mandibular lingula using CBCT. They reported that nodular, truncated, triangular, and assimilated shapes, which accounted for 51.2%, 32.0%, 14.1%, and 2.7 %, respectively.

  1. Materials and Methods:

The authors fail to mention how the sample size was calculated. Also have a sample size as 90 is very less for a conclusion they draw. Please justify.

Answer: In the paragraph of Materials and Methods (2.1. Data Conditions), we add 2 sentences “  We applied GPower version 3.1.9.2 (Franz, Universitat Kiel, Germany) to calculate sample size. After specifying α=0.05, we recruited 90 patients to achieve sufficient power of 83%.”

  1. The parameters with which the CBCT scanning was done is missing in the methodology. Please add it. The POV, Exposure time, etc to be mentioned is very important here.

Answer: In the paragraph of Materials and Methods (2.1. Data Conditions), we add 2 sentences:

During CBCT (New Tom VGi evo, Imola, Italy) scan, patients were under natural head position with centric occlusion. The sizes of the field of view (FOV) were 24x19cm. The CBCT was operated at 110 kV, 4.59 mA and X-ray emission time was 3.5 s with a voxel of 0.03 mm.

  1. The results and discussion is ok.

In Discussion it is suggested to the authors to add 2 review tables of previous studies in different populations and with the current investigation.

The review table can have the Authors, Year, Population, sample size, method use for evaluation (CBCT, OPG, Lat ceph, any other), dimensions

Answer: We add a new Table (Table 7) for literature review of lingula shape.

In literature review, there is no repost about antilingula shape.

Reviewer 2 Report

Dear authors,

Thanks for your kind submission and for letting me express my opinion, below you will find my suggestion and comments to improve the quality of your manuscript,

The introduction fails to provide a correct overview on this anatomical district, I suggest the authors to add consistent literature on the topic

M&M should include inclusion and exclusion criteria, also who performed the cephalometric tracing? Please see corresponding literature on 3D cephalometry: doi: 10.3390/bioengineering9050216
DOI: 10.1007/s00330-020-06905-7

Also the sample size should be analized, especially when assessing corretion between the skeletal class and the lingula anatomy,

Figure 2 shows segmented cts, please describe the procedure used,

Thanks

Author Response

  1. The introduction fails to provide a correct overview on this anatomical district, I suggest the authors to add consistent literature on the topic

Answer: We add the following sentences at the third paragraph of Introduction.

  • In India, Tuli et al. [9] firstly classified the lingula in the adult dry mandible into 4 shapes: triangular, truncated, nodular and assimilated shapes, which have prevalence rates of 68.5%, 15.8%, 10.9%, and 4.8%, respectively.
  • In Turkey, Sekerci and Sisman [17] identified morphological shapes of the mandibular lingula using CBCT. They reported that nodular, truncated, triangular, and assimilated shapes, which accounted for 51.2%, 32.0%, 14.1%, and 2.7 %, respectively.

  1. M&M should include inclusion and exclusion criteria,

Answer: In the 2. Materials and Methods  2.1. Data Conditions

We add the inclusion and exclusion criteria.

The inclusion criteria were as follows: (1) All patients were Taiwanese (2) the age of the patient was 16 years old or older (3) mandibles were dentate and included second molar (4) CBCT image showed the good contrast resolution quality for the classifica-tions of lingula and antilingula. The exclusion criteria were as follows: (1) patient was under 16 years of age, (2) mandibular pathology, (3) a history of orthognathic surgery, and (4) mandibular trauma.

  1. also who performed the cephalometric tracing? Please see corresponding literature on 3D cephalometry: doi: 10.3390/bioengineering9050216 and DOI: 10.1007/s00330-020-06905-7

Answer: (1) In the last sentence of Materials and Methods (2.1. Data Conditions), we add a sentence “Cephalometric analysis was performed by Chen CM.”

(2) In the third paragraph, we cite 2 references (15,16) as recommendation:

Reference

  1. Farronato M, Maspero C, Abate A, Grippaudo C, Connelly ST, Tartaglia GM 3D cephalometry on reduced FOV CBCT: skeletal class assessment through AF-BF on Frankfurt plane-validity and reliability through comparison with 2D measurements. Eur Radiol. 2020,30,6295-302.
  2. Farronato M, Baselli G, Baldini B, Favia G, Tartaglia GM. 3D Cephalometric Normality Range: Auto Contractive Maps (ACM) Analysis in Selected Caucasian Skeletal Class I Age Groups. Bioengineering (Basel). 2022,9,216.

  1. Also the sample size should be analized, especially when assessing corretion between the skeletal class and the lingula anatomy,

Answer: In the paragraph of Materials and Methods (2.1. Data Conditions), we add 2 sentences “  We applied GPower version 3.1.9.2 (Franz, Universitat Kiel, Germany) to calculate sample size. After specifying α=0.05, we recruited 90 patients to achieve sufficient power of 83%.”

  1. Figure 2 shows segmented cts, please describe the procedure used

Answer: The legends of Figure 1 and Figure 2 are revised.

Figure 1. In the medial aspect of right mandibular ramus, four types of lingula were classified. (A) Triangular (B) Truncated (C) Nodular (D) Assimilated

Figure 2. In the lateral aspect of right mandibular ramus, four types of antilingula were classified. (A) Hill (B) Ridge (C) Plateau (D) Plain.

Round 2

Reviewer 1 Report

Thank you for addressing to most of the comments 

It has surely made the flow of the paper more elaborative

Author Response

It has surely made the flow of the paper more elaborative

Answer:  We would like to thank the reviewers for their thoughtful comments and efforts towards improving our manuscript.

Reviewer 2 Report

Dear authors, thanks for your extensive revision and for following my advices,

Please a few minor revision are needed in my opinion before further revision:

3. Answer: (1) In the last sentence of Materials and Methods (2.1. Data Conditions), we add a sentence “Cephalometric analysis was performed by Chen CM.”

Rather than the name of the operator I would describe the level of experience: for example: "...was performed by an experienced operator with more than ... years in ...", Also it could be useful to assess inter operator variability on a small sample if possible.

5. segmentation is a process that allows to convert a CT's dicom into .stl, figure 1 diplays a section of the mandible's .stl and 2 is the full .stl, if this process was automatic please add the software used and the function.

Thanks

Author Response

  1. Answer: (1) In the last sentence of Materials and Methods (2.1. Data Conditions), we add a sentence “Cephalometric analysis was performed by Chen CM.”

Rather than the name of the operator I would describe the level of experience: for example: "...was performed by an experienced operator with more than ... years in ...", Also it could be useful to assess inter operator variability on a small sample if possible.

Answer: In the last sentence of Materials and Methods (2.1. Data Conditions), we add 2 sentences:

During 3D cephalometry on CBCT data, inter-operator variability was examined. The percent agreement (81/90) was 90 with a Kappa value of 0.85.

  1. segmentation is a process that allows to convert a CT's dicom into .stl, figure 1 diplays a section of the mandible's .stl and 2 is the full .stl, if this process was automatic please add the software used and the function.

Answer: In the 2. Materials and Methods (2.2. Image Investigation and Analysis) we add 2 sentences:

we add 2 sentences: “All CBCT images (DICOM files) were imported into RadiAnt DICOM Viewer (version 4.6.9, Medixant, Poznan, Poland) software to reconstruct and segment the 3D image. After extraction of the mandibular ramus (separated from the maxilla), four shapes of the lingula (Figure 1: triangular, truncated, nodular, and assimilated) [9] were observed.”
